# Bioenergy versus Soil Improvement: Policy Coherence and Implementation Gaps in Crop Residue-Based Bioenergy Development in China

**Jiaqi Zhang** [1,2,*] **, Peter Oosterveer** [2] **, Yu'e Li** [1] **and Mary Greene** [2]

1   Institute of Environment and Sustainable Development in Agriculture, Chinese Academy of Agricultural Sciences, Beijing 100081, China
2   Environmental Policy Group, Wageningen University and Reserach, 6706 KN Wageningen, The Netherlands
*   Correspondence: jiaqi.zhang@wur.nl

**Abstract:** Promoting crop residue-based bioenergy (CRB) is a strategy for mitigating climate change and ensuring water-energy-food security. Although China has abundant crop residues, CRB is not well developed, and several policy targets are not being reached. These implementation gaps need critical examination as CRB development mainly relies on policy support. This examination provides a framework to assess the feasibility of this policy that covers several cross-cutting issues. By reviewing national policies, this paper analyzes the horizontal coherence of multisectoral CRB policies during the last two decades. Based on 55 semi-structured interviews, conducted with stakeholders, civil servants and experts, we analyzed the vertical coherence among policy implementation to further explore the causes of the limited success of CRB policies. The priority given to the cultivation and livestock sectors (particularly for soil improvement) in crop residue utilization policies and the development of energy policy targets resulted in horizontal policy incoherence. Furthermore, financial limitations were shown to be the most significant obstacle to CRB policy implementation. Successful CRB policy implementation requires a debate about the crop residue utilization for either energy or fertilizer to determine the more sustainable application. Appropriate incentives and increased technological quality of bioenergy projects are also recommended.

**Keywords:** crop residue-based bioenergy; straw return; policy coherence analysis; multisectoral policy effects





## 1. Introduction

Increasing climate change and population growth raise the critical issue of food and energy security and related water resources management. Recent research estimated that globally 50% more energy [1] and 60% more food [2] will be required by 2050. This would lead to growing greenhouse gas (GHG) emissions and additional demands for arable land and water resources. At the same time, bioenergy is receiving significant attention in ensuring energy supply for two reasons. First, to meet the growth in energy demand, renewable energy could contribute 17–43% of the global energy supply by 2050, whereby bioenergy could potentially contribute some 100–300 EJ [3]. Second, contrary to the first-generation biofuels produced from food crops or energy crops grown on arable land, second-generation bioenergy produced from nonfood crops is expected to be an effective strategy to mitigate climate change. Second-generation bioenergy converts energy from the waste of food crops, agricultural residues, wood chips, and waste cooking oil without increasing the pressure on land resources, water usage from planting energy crops, or raising the GHG emissions from land-use change [4–6].

There is a growing interest in developing the potential of crop residue-based bioenergy (CRB), both technically and economically. CRB produced from agricultural byproducts is positioned as green energy, not hampered by the "food vs. fuel" debate. In theory, the

global potential of producing energy from agricultural residues from cereal and sugarcane cultivation is approximately 3.7 Pg, corresponding to 65 EJ/y accounts for 15% of the global primary energy consumption [7]. Researchers argue that the high costs for crop residue logistics [8] and CRB production [9] limit CRB development. Hence, CRB development depends heavily on subsidies. Research on the recent development of the CRB industry in China found that the institutional infrastructure, technical performance, financial support, and public adoption of CRB products and consumers' income are all critical [10,11]. Effective policies could overcome these shortcomings by applying adequate policy instruments but very little is known about the impact of current policies within the CRB sector.

Trade-offs between CRB and other crop residue applications are inevitable, but few studies have systematically analyzed them through a policy perspective. Crop residue incorporation and collection are the major ways to dispose of crop residues. Energy utilization, as well as livestock feed, CRB, mushroom cultivation, building materials, etc., reuse collected crop residues as sustainable raw materials [12,13]. Straw return (SR) provides ecosystem services, such as increasing crop yield and carbon sequestration [14–16]. Competition among different crop residue applications exist as the quantity of crop residues is limited. CRB is an industry intertwined with multiple sectoral policies. Agricultural policies affect the development of CRB, as its raw material is from agricultural byproducts so the CRB industry should not negatively affect agricultural production. This leads to a complex policy challenge. Two other important policy domains are environment and energy [17]. Policies from different domains have their concerns on water, energy or food resources. Thus, there is a danger that a lack of coherence may appear between the different policy domains and the different policy aims within them while their practical implementation may also show gaps compared with their original aims [18,19]. Analyzing policy coherence across these overlapping sectors and policy implementation gaps is therefore important.

Coherent policies could reinforce shared policy aims across different government departments and avoid or minimize negative spillovers [20,21]. Policy coherence analysis studies the relationships between different policy domains [22] as well as the instruments and implementation of policies. Policy coherence has different dimensions, including internal (within a policy area), vertical (between different levels of governance), horizontal (with other sectoral policies), and transnational (between national and international policies) [22]. According to Nilsson et al. [23], despite policies being coherent in terms of their objectives, incoherence often occurs with respect to policy instruments and policy implementation practices. For instance, studies on bioeconomy showed the presence of trade-offs between bioeconomy and agro-food policies, even though there is coherence at the level of policy goals [24]. Further analysis is needed across different policy topics and domains to shed light on the role of different policies and their interactions across levels and between directly and indirectly involved sectors.

Policy coherence analysis is widely used to assess combined policy effects in water-energy-food nexus issues. Several attempts have been made to analyze policy coherence across water-energy-food sectors [25,26]. For example, analysis between climate and forestry indicates that economic factors affect the extent of logging harvest and bioenergy development [27], whereas policy-based legislation and supportive collaborative multi-stakeholder approaches are effective for addressing climate change adaptation [28]. The importance of taking an actor-focused perspective was emphasized when assessing economic issues, and public interest and adoption in policy implementation [29,30]. Thus far, however, coherence in CRB policies has hardly been studied. Therefore, this paper intends to fill this gap and thereby contribute to a more effective and efficient development of the CRB sector.

This study seeks to make an empirically informed contribution to this field by analyzing the horizontal coherence of national multisectoral policies, including agricultural, energy and environmental scopes, on CRB performance in China and the vertical coherence among policy implementations. Focusing on the provinces of Heilongjiang and Hunan, we

applied a qualitative approach to investigate the influence of potential conflicts between national and local policies, and multisectoral policies on the implementation, outcomes, and effectiveness of these multisectoral policies. This paper firstly intends to explain the degree of horizontal policy coherence across water-energy-food policies in these sectors and the vertical incoherence in their implementation in China's CRB sector. In doing so, it seeks to contribute to a better understanding of the problems arising from trade-offs that occur within multisectoral management in reaching the targets in bioenergy development globally. Furthermore, our study may advance policy analysis scholarship and, in particular, understanding of policy coherence across horizontal and vertical dimensions [22,28] on the rationality and feasibility of ongoing policies in nexus issues, especially for complicated issues with many trade-offs.

This paper is structured as follows. The section "Crop residue utilization in China" presents the necessary brief background of the introduction and promotion of CRB as part of the crop residue reuse policy in China. The section "Method and materials" describes the methodology we used to analyze policy coherence. The "Results" section presents the horizontal and vertical coherence analysis based on policy review and a semi-structured interview. The "Discussion" section highlights the horizontal coherence at the national policy level and the vertical incoherence of policy implementation in Heilongjiang and Hunan, and explores several policy recommendations for CRB's future development and the comprehensive utilization of crop residues. Finally, all findings are summarized in Section 5 where we also formulate our conclusions.

## 2. Crop Residue Utilization in China

Crop residue utilization is cross-cutting and included in different policy domains and is therefore influenced by multiple policies and actors. In China, the Ministry of Agriculture and Rural Affairs (MARA) is leading in crop residue utilization. Other governing institutions include the State Council (SC, the chief administrative authority of China, and ministries which are the branch departments for different sectors within the SC), the Ministry of Finance (MOF), the National Development and Reform Commission (NDRC), and the Ministry of Ecology and Environment (MEE). In addition to the above sectors, the CRB development is under the governance of the National Energy Administration. The decentralization strategy in China, leading to more flexible local governance arrangements, may result in a divergence between national and local environmental policies. Moreover, the market-oriented growth reforms of infrastructure projects increase economic competitiveness [31]. This flexibility has resulted in new policy instruments and policy actors, but they may also increase the uncertainty of policy outcomes [32].

In the context of China, there are five possible applications for the utilization of crop residues: straw return (SR), animal forage, CRB, substrate for mushroom cultivation, and industrial feedstock as a replacement for wood [12]. Since the prohibition of open-air straw burning, treating and disposing the large amounts of crop residues has become a major issue for the Chinese government [33]. Especially after 2007, when the Chinese government introduced ecological civilization as a policy goal and emphasized forming "an energy- and resource-efficient and environmentally friendly structure of industries, pattern of growth and mode of consumption" [34], the shift to a circular economy and resource recycling has become important. The government has implemented various policies to promote crop residue reuse. The amount of reused crop residues has doubled and SR and CRB use was sharply boosted. Of all five applications for utilizing crop residues, SR, forage, and CRB are most used. These three applications accounted for more than 80% of use in 2018 (as shown in Figure 1).

The government promotes SR by providing financial support to purchase crop residue crushing and returning machines to implement SR and to offer information and training. As a result, SR plays an overwhelming role in crop residue utilization, with a marked increasing share of collectible crop residue from 9.8% in 2005 to 57.6% in 2018. According to the monitoring data from 32 sites located in important agricultural production zones,

the soil organic matter and crop yield increased by 5–7% and 2–4.5%, respectively [35]. However, contradictory findings about crop residue incorporation have also emerged, including about the marginal benefits of soil improvement [36] and insect problems [37].

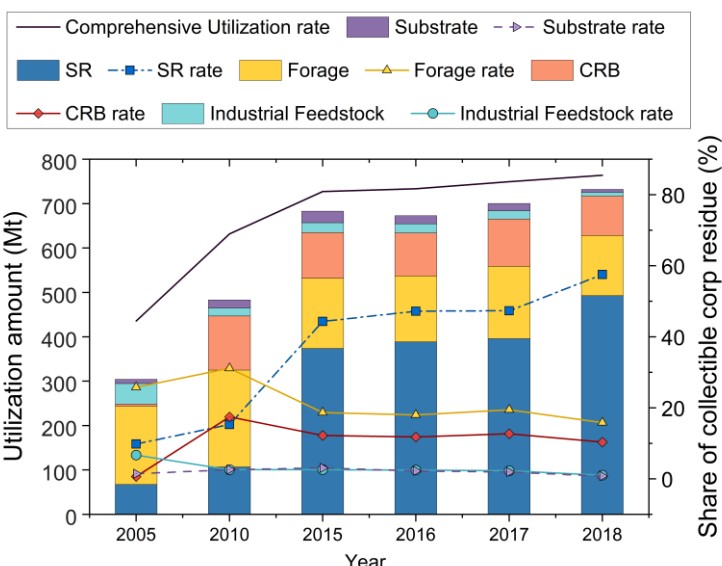

**Figure 1.** The development of comprehensive utilization of crop residues [36–38].

Different from the smooth promotion of SR, China's CRB strategy has not developed as expected even though there is a significant amount of raw material available (theoretically more than 800 million tons annually [39]) and an encouraging policy context exists. Building on the increasing demand for energy and climate change mitigation, the Chinese government has boosted bioenergy projects through direct subsidies for bioenergy products and tax reductions for manufacturers since 2006 [40,41]. However, according to the data on CRB [41–43], except for straw pellet and biodiesel between 2006 and 2010, the development did not achieve the targets (as Figure 2). Meanwhile, China announced its aim to have a $CO_2$ emissions peak before 2030 and to achieve carbon neutrality before 2060 [44]. To ensure carbon neutrality, the Chinese government has announced that "non-fossil fuels in primary energy consumption would account for 25 percent by 2030" [45]. This requires an effective and coherent multisectoral policy also in China's CRB sector.

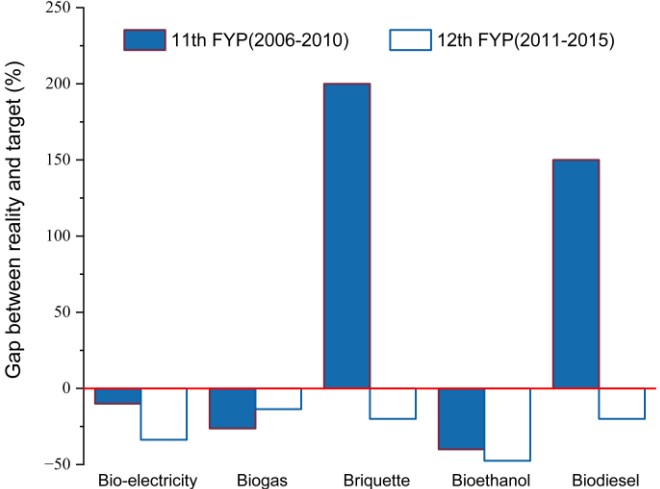

**Figure 2.** The gaps between policy targets and the reality of CRB development in China. The red line (0%) represents the actual CRB products meeting the policy targets. The columns above and below the red line indicate that the amount of CRB that is beyond or below the policy targets [41–43].

## 3. Materials and Methods

As CRB is a water-energy-food system, the sectors under investigation include agricultural, energy and environmental policies related to crop residue recycling. In this study, we applied a qualitative nexus approach for an in-depth assessment of policy coherence (horizontal coherence analysis) and policy implementation practices (vertical coherence analysis) [29,46]. Our analysis of horizontal coherence focused on the national policy documents governing CRB and other applications of crop residues. The vertical coherence analysis was conducted in the provinces of Heilongjianng and Hunan.

### 3.1. Horizontal Coherence Analysis

#### 3.1.1. Policy Retrieval

We followed a structured, multistage process to systematically identify and analyze the CRB policy documents. National policies relating to crop residues were collected and screened, as policies promoting comprehensive utilization of crop residues have impacts on CRB development. Policy documents were collected through keyword searches from the relevant ministries (as shown in Table 1). We used "crop residues", "straw", "renewable energy," and "bioenergy" as search terms. All four of these keywords were searched in each relevant ministry. More than 700 policy documents were collected from official websites, which included websites for China's state council, the environmental sector, the energy sector, and the agricultural sector. The search was further refined to identify important policy documents. First, the sampling period for selecting policies ranged from 2000 to 2020 because policies implemented before 1999 only had guiding ideas but did not have specific actions. Furthermore, the implementation of the Measures for Straw Burning Ban and the Comprehensive Utilization of Crop Residues (MEE 1999-98) clearly banned straw burning and provided policy support for the comprehensive utilization of crop residues. Second, we only focused on policies with specific instruments affecting the comprehensive utilization of crop residues, including CRB and soil improvement.

**Table 1.** Keywords and ministries.

| Keywords | Search Ministries |
|---|---|
| Straw; crop residues; renewable energy; bioenergy | The State Council (SC), PRC<br>Ministry of Agriculture and Rural Affairs (MARA), PRC<br>Ministry of Ecology and Environment (MEE), PRC<br>National Development and Reform Commission (NDRC, including National Energy Administration, NEA), PRC |

Note: SC: https://www.gov.cn/ accessed on 11 August 2020; MARA: http://www.moa.gov.cn/ accessed on 14 August 2020, formerly named Ministry of Agriculture; MEE: http://www.mee.gov.cn/ accessed on 6 September 2020; NDRC: http://www.ndrc.gov.cn/ accessed on 10 September 2020.

We selected 70 national policies with clear objectives or specific instruments from 2000 to 2020, of which 47 policies are still effective and 23 have expired. According to their aims, these policies can be divided into three categories: green agricultural development, environmental protection and resource recycling, and renewable energy. Further detailed information, including name, document number, and responsible ministry, is shown in Appendix A.

#### 3.1.2. Screening for Policy Instruments

Our horizontal coherence analysis was based on the method by Nilsson et al. [23]. First, an inventory was made of all effective national policy instruments. These instruments were classified into regulatory, economic, informative instruments and government provision of public goods or services [47]. Each category of instruments is based on a different rationale regarding the way actors are steered: by restricting or allowing behavioral options (regulatory instruments), by changing the cost-to-benefit ratios of these options (economic instruments), by informing about different options (informative instruments) [48–50], or by providing and planning infrastructure and other public resources and building capacity

within the government (government provision of public goods or services) [47]. Second, the impacts on CRB or other crop residue applications were coded in Atlas.ti and classified into three categories: (1) promoting and maintaining the development of CRB or other applications (+), (2) weakening the development of any application (−), and (3) unknown impacts on certain applications (?). Based on the policy impacts on single applications, synergies and conflicts, these instruments were classified as mutual benefits (++), mutual loss (−−), or trade-offs (+− or −+) (as shown in Figure 3). Policy instruments are coherent if the impacts on different applications are mutually beneficial or detrimental. Finally, a systematic analysis of policy coherence was conducted to determine the trade-offs between CRB and other applications of crop residues and to determine the effective instruments for different applications.

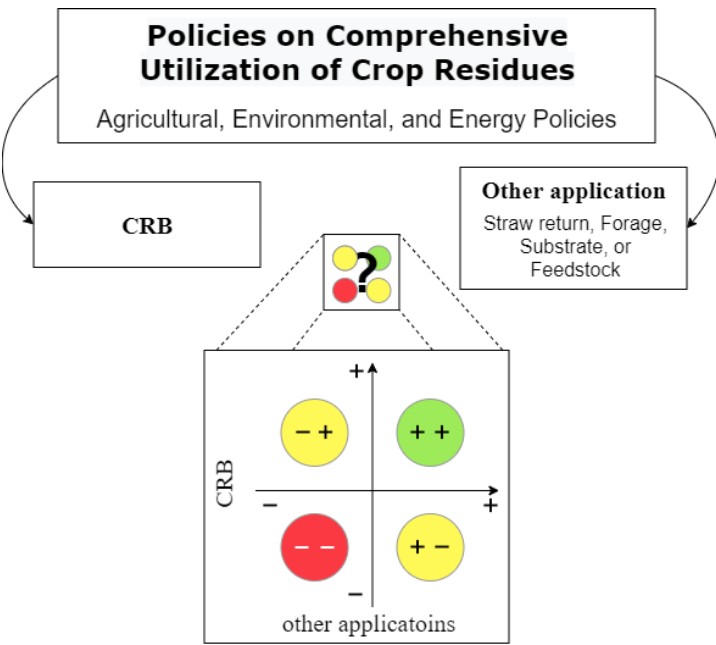

**Figure 3.** Illustration of coherence analysis. (adapted from Makkonen et al. [46]).

### 3.2. Vertical Coherence Analysis

To assess vertical coherence, interviews were conducted in Heilongjiang and Hunan Provinces because these two provinces are the main grain-producing areas in China, with different crops, climate conditions, cropping systems, and crop residue applications. The different cropping systems result in different implementation practices of national policies and special provincial policies. The provincial policies relating to crop residues were collected through document review and interviews with local civil servants and used for vertical coherence analysis (shown in Table 2). The policies recommended by the local officials complemented the already selected policies, such as the HLJ 2018-39 only implemented in pilot regions. Both the effectiveness of the policy outputs and the interviewees' requirements or recommendations are further discussed in the vertical coherence analysis.

**Table 2.** Provincial policies that influence the comprehensive utilization of crop residues.

| Policy Area and Reference | Policy | Status |
|---|---|---|
| Heilongjiang Province | | |
| HLJ 2016-113 | The Implementation Plan for Prohibiting Straw Incineration to Improve Air Quality in Heilongjiang Province | Effective |
| HLJ 2018-39 | The Three-Year Action Plan for the Comprehensive Utilization of Crop Residues in Harbin, Suihua, Zhaozhou and Zhaoyuan in Heilongjiang Province | Effective |
| HLJ 2019-16 | The Implementation Plan for the Comprehensive Utilization of Crop Residues in Heilongjiang Province in 2016 | Expired |
| HLJ 2020-18 | The Implementation Plan for the Comprehensive Utilization of Crop Residues in Heilongjiang Province in 2020 | Effective |
| Hunan Province | | |
| HN 2015-28 | The policies in Hunan Province include The Opinion on Strengthening Rural Energy | Effective |
| HN 2018 | The 13th five-year Plan for bioenergy in Hunan Province | Effective |
| HN 2018-84 | The Implementation Opinion of the Innovations in Systems and Mechanisms for Green Agricultural Development in Hunan Province | Expired |

A total of 55 semi-structured interviews were conducted with stakeholders (i.e., farmers, middlemen and manufacturers), local civil servants and experts (the exact number of each kind of interviewee is listed in Table 3). Farmers came from eight different main grain-producing counties across the two provinces in each province. According to the local cropping system, we selected corn and bean farmers in Heilongjiang Province and rice farmers in Hunan Province. Both family-scale and cooperative farms were included. Four of the six county-level servants and one provincial civil servant from Heilongjiang Province were included, as well as two county-level and one provincial civil servants from Hunan Province.

**Table 3.** List of stakeholders, local officers and experts interviewed.

| Informant | Position (and Number of Respondents) | Interview Method |
|---|---|---|
| Stakeholders | (44) | |
| Bean farmers | (2) | Face-to-face |
| Corn farmers | Family-scale farms (15), cooperative farms (6) | Face-to-face |
| Rice farmers | Family-scale farms (10), cooperative farms (4) | Face-to-face |
| Logistics companies | (4) | Face-to-face |
| Manufacturers | (3) | Face-to-face/phone interview |
| Local civil servants | County level (6), provincial level (2) | Face-to-face |
| Experts | Comprehensive utilization of crop residues (1), climate change (1), power generation (1) | Face-to-face/phone interview |

Each interview lasted approximately 30 min and was conducted either face-to-face or by phone. Interviews were recorded by audio recording or notes. Stakeholders were interviewed about the national and provincial policy outputs with respect to crop residue reuse. Local civil servants and experts were interviewed to give a broader perspective and to reflect on their overall experience during policy implementation, such as the efficiency of policy outputs, restrictions, and recommendations. Based on these interviews, we mapped the practices and effectiveness of different policy outputs and further identified implementation gaps.

## 4. Results and Discussion

This section presents the results from our analysis of the horizontal and vertical coherence of CRB policies in China. Horizontal coherence is analyzed by mapping policy instruments. The results are further discussed to determine the incoherence in CRB policies that could impede CRB development. Vertical coherence analysis is based on interviews with stakeholders, civil servants, and experts. Their feedback on policy implementation is summarized and further discussed to determine the incoherence and gaps in CRB policies, especially with respect to policy implementation.

### 4.1. Governance of China's CRB

The policies affecting CRB and crop residue utilization are formulated and implemented by several ministries in China. Managing the ban on burning crop residues to improve air quality was a critical point until 2007, when the ambition for Ecological Civilization was included in China's governmental documents [34]. The MEE in its policy for 1999-98 was the first to prohibit the burning of crop residues and recommended CRB, forage, and industrial feedstock to dispose of crop residues, but only few policy instruments effectively boosted CRB development. In this period, MEE was in charge of governing crop residue disposal using regulatory and informative instruments. Prohibition of burning straw was included in the Atmospheric Pollution Prevention and Control Law (version 2000). In addition, there was active information campaigning aimed at the general public, especially farmers, to highlight the harm from burning straw.

Along with sustainable development moving into the spotlight of societal development, the core of crop residue disposal policies shifted from managing the burning ban to their comprehensive utilization. Figure 4 shows the integration of CRB policies in multiple sectors, their validity, and the types of policy instruments. The implementation of the Opinion on Accelerating the Comprehensive Utilization of Crop Residues (SC 2008-105) marked the moment when the policy goal became the comprehensive utilization of crop residues [51]. An increasing number of policies from multiple sectors and the SC aims at resource recycling, agricultural development, and renewable energy contributing to crop residue utilization. The MARA takes the overall responsibility for crop residue utilization. Economic instruments (e.g., subsidies and tax reductions) and the government public goods and services provision (e.g., biogas and heat pipe) contribute to CRB development. All four kinds of instruments have had impacts on the crop residue sector. Nevertheless, the Implementation Plan for Comprehensive Utilization of Crop Residues during the 12th Five-Year–the first five-year plan aiming to comprehensively utilize crop residue–indicated that crop residues should be mainly used for agricultural production, including planting and livestock, as crop residues are agricultural byproducts.

The governance of local crop residues was supervised and financed by national and local governments. National policies are devolved to provincial levels by pilot programs and target responsibility systems, which are rules governing job assignment, performance appraisal, and remuneration. These pilot programs could receive special funds, from national and local sources, after approval and meeting an acceptance review. Within target responsibility systems, national targets are assigned to provincial levels. The evaluation results of meeting these targets are included in the annual evaluation of provincial officials. In addition, significant investment approvals (including the budget from the central government and pilot programs) are withheld to punish provinces that fail to achieve the targets [52,53].

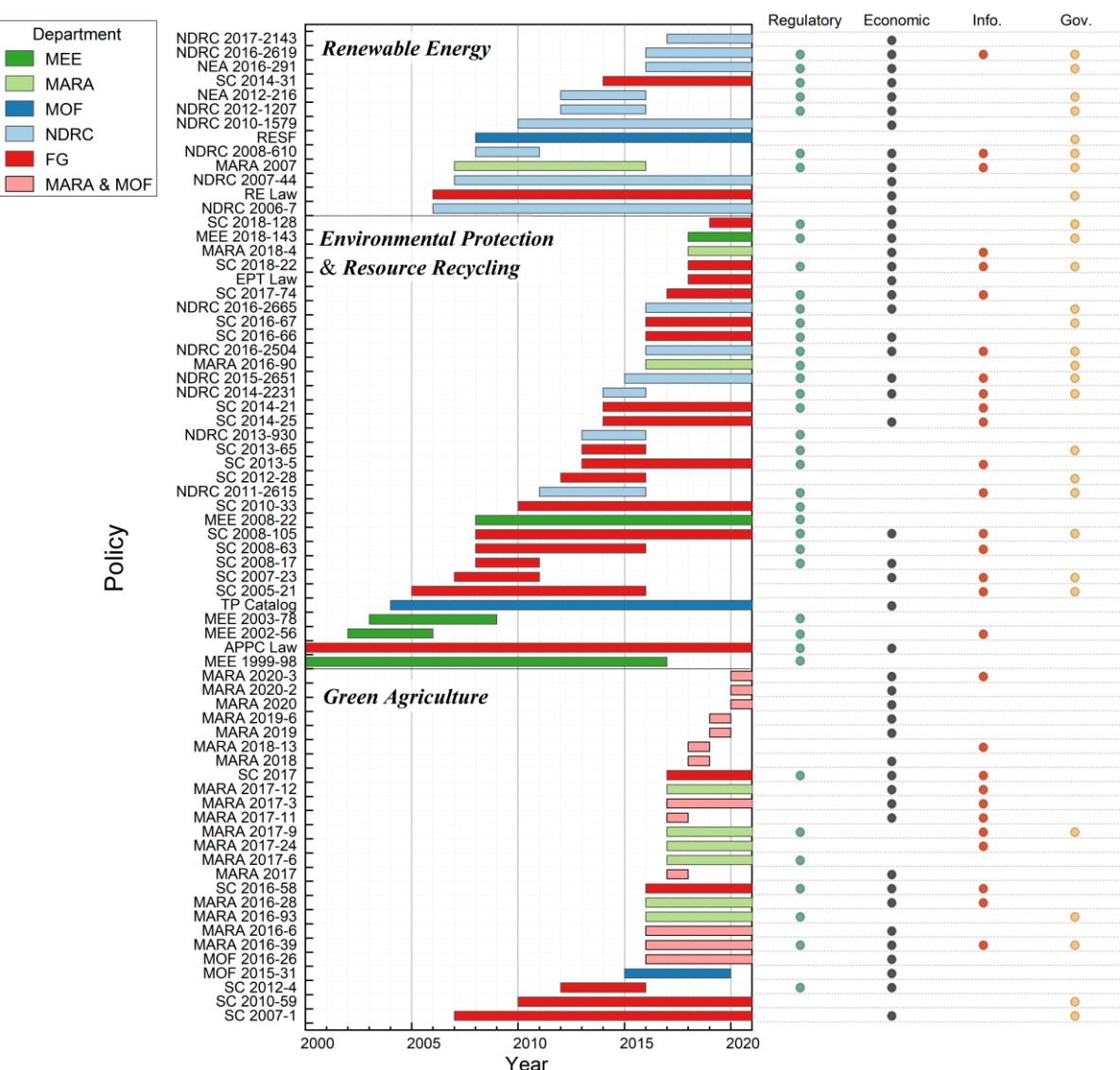

**Figure 4.** China's national policies concerning crop residues. FGs are the policies governed by the SC and/or more than three departments. Gov. denotes the government public goods and services provision. The policy documents for each abbreviation are shown in Table A1 in Appendix A.

*4.2. Hovizontal Coherence Analysis*

4.2.1. Horizontal Coherence among National Policies

We found no policies specifically promoting the development of crop residue-based forage, substrate, and industrial feedstock. The policies promoting these three applications are contained within the policies promoting all applications for utilizing crop residues. Therefore, we classified the instruments depending on the crop residue utilization process and showed their policy impacts on CRB, SR, and the three other applications, but focused on the coherence between CRB and SR. As shown in Figure 5, more policies, including positive, negative and unknown instruments, are linked to green agriculture and environmental protection & resource recycling. In contrast, the comprehensive results involving all policy instruments of CRB indicate a better performance of all policy sectors, especially with a prominent positive result (i.e., 100%) of energy policies.

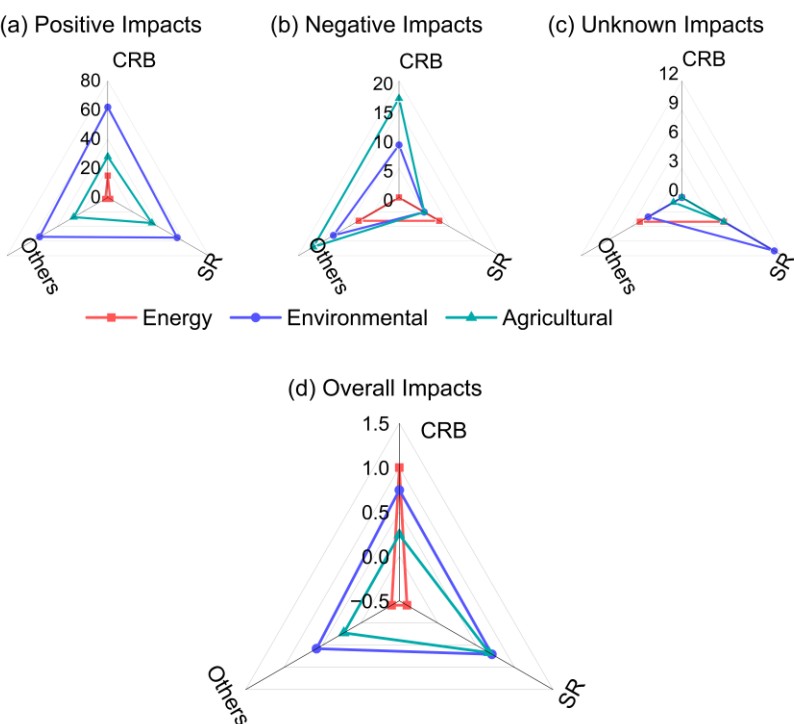

**Figure 5.** The instruments' impact of effective policies on CRB, SR and other CR applications. (**a**–**c**) The numbers of policies with positive (+), negative (−) and unknown (?) instruments on each different CR application. (**d**) The comprehensive impact of all effective policies by assuming positive, negative and unknown impacts indicate +1, −1 and 0, respectively, and calculating the shares of policies from each sector. Note, each policy could have more than one instrument, so that the summation is more than the total number of all effective policies.

The analysis indicates that mutual benefits and mutual losses only occur in crop residue production and in general policies that contain all five applications (See shown in Table 4, more details about the policies related to each instrument can be found in Appendix B). Overall, these policies targeted the early part of crop residue processing, which has mutual benefits (prohibiting straw burning) and mutual losses (fallow) to all five applications. Fallowing leads to a decrease in the amount of crop residues, with common losses in CRB and other applications. MARA 2018-4, SC 2017, SC 2016-66, MARA 2016-6, MARA 2016-28, and MARA 2020-3 promoted fallowing to increase soil fertility by designating particular areas and financing the farmers applying fallowing for three to five years. In contrast, mutual benefits, i.e., win-win effects on all applications, have been generated by policy instruments related to prohibiting straw burning and promoting crop residue utilization. In the case of the prohibition of burning straw, it was necessary to find greenways for disposing the crop residues that were previously burned in the open air. Therefore, the amounts available for crop residue utilization increased indirectly. The latest target for utilizing crop residues was increasing the amount with 85% by 2020. Subsidies and tax reductions to agents who reuse crop residues have direct benefits for all applications. Moreover, the environmental tax and discharge fees for companies responsible for large pollution effects, such as coal power and chemical fertilizer plants, indirectly promote all applications for utilizing crop residues.

Table 4. Policy Impacts on Crop Residue Utilization.

| Policy Outputs | Type | CRB | SR | Others |
|---|---|---|---|---|
| **CROP RESIDUE PRODUCTION** | | | | |
| Financing fallow for sustainable agricultural development | Eco. | (−) | (−) | (−) |
| Prohibition on burning straw in the open air, including fines, information, developing regulations, etc. | Reg. | (+) | (+) | (+) |
| **CROP RESIDUE LOGISTICS** | | | | |
| Encourage private sectors to engage in crop residue logistics by financing the construction of crop residue infrastructure, i.e., warehouse and land use | Gov. Services | (+) | (?) | (+) |
| Financing the collection and transport of crop residues from arable land | Eco. | (+) | (?) | (+) |
| **COMPREHENSIVE UTILIZATION OF CROP RESIDUES** | | | | |
| Increasing the overall utilization ratio of crop residues | Reg. | + | + | + |
| Financing innovations in manufacturing equipment to increase efficiency in crop residue processing | Gov. Services | + | + | + |
| Encourage public participation by involving the public in the implementation and assessment of policies | Info. | + | + | + |
| Promoting utilization by providing technical support | Info. | + | + | + |
| Financing the applications for crop residue utilization, including subsidies and tax reductions | Eco. | + | + | + |
| Environmental tax, or decreasing financial support for fossil fuel-based products/chemical fertilizer | Eco. | (+) | (+) | (+) |
| Crop Residue Utilization for Bioenergy | | | | |
| More bioenergy/CRB is needed to reach the GHG emission reduction target | Reg. | + | (−) | (−) |
| Financing CRB producers, i.e., tax reduction, subsidies for factory construction, and guaranteeing the purchase of CRB products | Reg. & Eco. | + | (−) | (−) |
| Constructing infrastructures for transporting CRB products, i.e., power grid and biogas pipes | Gov. Services | + | (?) | (?) |
| Financing CRB users, e.g., subsidy for straw pellet stoves or for switching from coal to straw pellet, etc. | Eco. | + | (?) | (?) |
| Crop Residue Utilization for Soil Improvement | | | | |
| Crop residues should be returned to arable land to increase soil fertility | Reg. | (−) | + | (−) |
| Financing SR to arable land, including subsidies for purchasing agricultural machines and SR | Eco. | (−) | + | (−) |

Note: "+" strengthening or positive impacts, "−" weakening or negative impacts, and "?" unknown impacts that could be positive, negative or neutral impacts. The parentheses in the effects indicate that the effect of the output is indirect. Eco. denotes economic instruments, Reg. denotes regulatory instruments, Info. denotes informative instruments, Gov. Services denote the government public goods and services provision.

Policies focusing on SR and CRB have directly and indirectly generated trade-offs. Targets of 15% non-fossil fuel use in primary energy consumption and a CRB consumption equivalent to 58 million tons of standard coal by 2020 have been integrated into the 13th Five-Year Plan of energy development. On the one hand, in addition to tax relief for CRB production, the government asks power transmission companies to buy all CRB-based bioelectricity at a subsidized price that is higher than for coal-based power if the CRB power plants are approved. On the other hand, consumers of CRB products, mainly using straw pellets, can receive subsidies to purchase and install straw pellet stoves or switch from coal to straw pellets. Moreover, national and local governments develop infrastructures to promote the consumption of CRB products in rural areas, such as heat and gas pipes. On the other hand, the government has also promoted straw reuse to increase soil fertility by offering subsidies and assistance in purchasing specialized machines. The competition for raw materials generated trade-offs in crop residue utilization.

Finally, we identified four policy instruments that could not be classified into mutual benefits, mutual losses, or trade-offs. These instruments, two of which belong to crop residue logistics and the other two to developing CRB infrastructure, have unknown impacts on the different applications. The logistics of crop residues have unknown effects

on fertilizer utilization but result in positive effects on CRB and the other three applications. Otherwise, agricultural machines crush crop residues and blend them with the soil in the field; hence, logistics has little impact on SR. Similar to the economic instruments for SR and CRB, the government has provided financial support to collect and transfer crop residues and to develop infrastructure, such as warehouses. In addition, to meet the CRB targets, the government invests in CRB infrastructure, including the power grid and biogas pipelines. This helps CRB sellers transfer products to consumers. Because finances for infrastructure were given after or simultaneously with the CRB projects, they have few impacts on allocating crop residues or changing the development of other applications.

4.2.2. Policies Are Horizontally Coherent in Promoting CRB Development

Our analysis of policies addressing the five applications for utilizing crop residues in China shows that most policy instruments prefer CRB, even though some policies for crop residue utilization (e.g., NDRC 2011-2615) give priority to the cultivation and livestock sectors. These results are consistent with previous research on policy preferences in promoting multiple ecosystem services whereby the option with a market value has priority over others [46,54]. Even though reusing straw for soil fertility is a form of carbon sequestration, a market for carbon sequestration has not yet been introduced in China, and therefore this has no market value yet. Carbon trading only functions at a small scale through some pilots [55]. Moreover, observing the effects of using straw for improving soil quality and enhancing crop yield takes considerable time and therefore does not have quick economic benefits.

Ideas about recycling crop residues have been integrated into the agricultural, environmental, and energy sectors. Agricultural policies promoting crop residue utilization preferred reusing straw for soil quality, while the policies formulated in the environmental and energy sectors preferred CRB. However, when considering all policies related to crop residues, more effective instruments were focused on CRB rather than on SR (as shown in Table 5). Policy instruments that supported the entire process of CRB production and consumption included factory construction, crop residue logistics, production, and product distribution. Not only the number of policy instruments with a focus on CRB, but also the scale of CRB showed the effectiveness of these policies. As shown in Figure 1, the share of CRB in collectible crop residues increased with almost the same amount as the share of crop residue utilization in the collectible crop residues from 2005 to 2010, while the share of SR increased only slightly. A total of 65.7% of the increase in crop residue utilization came from CRB development during this period (see Figure 1).

**Table 5.** Summary of the coherence analysis results (based on Table 4).

| | CRB | | | SR | | | Others | | |
|---|---|---|---|---|---|---|---|---|---|
| | Direct | Indirect | Total | Direct | Indirect | Total | Direct | Indirect | Total |
| + | 9 | 4 | 13 | 7 | 2 | 9 | 5 | 4 | 9 |
| − | 0 | 3 | 3 | 0 | 3 | 3 | 0 | 5 | 5 |
| ? | 0 | 0 | 0 | 0 | 4 | 4 | 0 | 2 | 2 |

The coincidence of rapid CRB development and CRB policy implementation has two main causes. First, economic instruments, including subsidies for crop residue collection and tax relief for CRB production, effectively boosted CRB production. Studies have shown the higher costs for producing CRB, especially the costs for collecting and processing crop residues, compared with traditional energy products based on fossil fuel [7,56]. The purchase of subsidized CRB bioelectricity dealt with the concerns about product sales. Second, financing the construction of the power grid and biogas pipes connected CRB to the established energy market. Other public benefits were gained through applying a market orientation. For example, new employment opportunities and additional energy supply encouraged public policies supporting CRB. This finding broadly supports the

work of other studies that demonstrated the importance of coherent multilevel policies in achieving the policy target [57]. This study shows that a horizontal coherence analysis of the policies from all relevant sectors sheds light on the impacts of multisectoral policies and allows for further developing better policies.

### 4.3. Vertical Coherence Analysis

4.3.1. Vertical Coherence among Policy Implementations

In this section, we focus on the practical implementation of CRB policies on the ground and evaluate the policy incoherence and relevant gaps therein. Based on information about the provincial policies provided by local civil servants, there are no direct conflicts with national policies in Heilongjiang and Hunan provinces, but there are different preferences in each province. In both provinces, all regulatory, economic and informative instruments and the government public goods and services provision have been directly and indirectly applied to promote the utilization of crop residues. The interviewees' comments and suggestions with respect to these policy instruments are summarized in Table 6. In general, all interviewees recognized the effectiveness of crop residue utilization policy instruments but indicated different requirements and suggestions for further developing policy support for CRB and other applications.

**Table 6.** Summary of findings from stakeholder interviews.

| Interviewees | Response to the Question | |
| --- | --- | --- |
| | **1. How Do Policies Work?** | **2. Policy Requirements and Suggestions** |
| Farmers/consumers of straw pellet | • Subsidies for straw stoves and straw pellets help develop CRB (only in some areas). | • The heat value of straw pellet is lower than that of coal but has a similar price; therefore, a subsidy for straw pellets is necessary to boost the use of straw pellets.<br>• The household heating system needs to be changed to suit straw pellets. Therefore, if the subsidy on purchasing straw stoves can be promoted widely, it will be helpful. |
| Corn Farmers | • Policies are effective in prohibiting burning straw.<br>• Policies work well in promoting SR and CRB. | • Subsidies for collecting and baling crop residues are not sufficient, and therefore farmers need to pay; therefore, more subsidies or new technologies that can lower prices are needed. |
| Bean Farmers | • Policies are effective in prohibiting burning straw.<br>• Policies work well in promoting SR and CRB. | • Government will change and ban the direct use of unprocessed beanstalks as fuel at home. |
| Rice Farmers | • Policies are effective in prohibiting burning straw.<br>• Compared with CRB, SR is suitable for rice straw; therefore, CRB policies have little impact on promoting CRB. | • Subsidy for reusing rice straw to improve arable land is not sufficient, and farmers need to pay; therefore, more subsidies or new technologies that can lower prices are needed. |

**Table 6.** *Cont.*

| Interviewees | Response to the Question | |
|---|---|---|
| | **1. How Do Policies Work?** | **2. Policy Requirements and Suggestions** |
| Logistics Companies | • Policies are good at collecting and reusing crop residues.<br>• Subsidies come from local finance; therefore, levels are different between provinces. | • National subsidies for collecting crop residues are only given in pilot areas. The government should increase the subsidy.<br>• Much crop residues are collected and baled by the cooperative renting out agricultural machines, and some of them lack information about the demand of processing companies. |
| Manufacturers | • Policies help promote crop residue utilization. | • Problems relating to the instability of crop residue prices and the logistics of crop residue application should be fixed.<br>• The approval process is complex and takes time; therefore, the subsidy is usually received only years after factory construction. Subsidies should be put in place on time.<br>• Technical innovations are needed to lower the production costs. |
| Civil Servants | • Policies work well in prohibiting burning straw in the open air.<br>• Policies help promote crop residue utilization. | • Straw applied for soil fertility is the first choice to reusing crop residues, as the cost is the lowest. More finance is needed for other applications when SR cannot absorb all crop residues.<br>• Local finances pay a lot in supervising burning straw and utilizing crop residues. With the decreasing fines received from burning straw, financial pressure would increase rapidly.<br>• After crop residue collection or straw reuse for soil fertility, farmers partly pay for soil preparation to plant next year, but low crop prices resulting in low income negatively affect technology adoption. |
| Experts | • Policies work well in promoting crop residue utilization. | • The CRB development depended heavily on subsidies, as raw materials account for much of the production costs. The unstable crop residue price plays a vital role in CRB development.<br>• Small-scale CRB, such as power and heat cogeneration and straw pellet, is suitable for rural areas if the production costs would go down.<br>• Technical innovations are needed to lower the production costs.<br>• Regulating the carbon market can balance the higher production costs for CRB and the environmental benefits. |

All groups of stakeholders considered the ban on burning straw in the open air the most effective measure followed by the policies promoting straw reuse for soil fertility and CRB. Regulations, such as the Atmospheric Pollution Prevention and Control Law of the People's Republic of China (APPC Law, 2000 version), restrict straw burning and regulate punishments. In the last two decades, the national government has increased the priority of environmental protection in responsibility contracts. To ensure the management of the ban on burning straw, local governments have given much attention and support to monitoring and informing the public about its negative impact on the environment. Especially after the Air Pollution Prevention and Control Action Plan and SC 2014-21, real-time air quality monitoring data is published; therefore, stakeholder groups of citizens and media can easily be informed and exert social pressure on local governments and on the people who burn crop residues.

The interviewed stakeholders mentioned different aspects of crop residues utilization policies, and farmers planting different crops had further comments. Beanstalks are usually collected and used as household fuel and forage in Northeast China; moreover, the small quantity and high fertility of beanstalks has always encouraged farmers to return them into the soil. Therefore, the crop residues utilization policies have little direct impact on their use. However, corn and rice farmers are affected by crop residues utilization policies. Because the straw-to-grain ratio is larger for corn than for rice, this results in more crop residues for corn farmers, who then need to take more efforts to dispose of corn stover. Farmers and their families are suppliers of crop residues but also CRB product consumers, especially by using straw pellets. The subsidy for consuming straw pellets and the information campaigns for using them encouraged people to adopt CRB. Economic support is used to promote straw pellets because of its lower heating value, short combustion time and the financial costs implied in modifying stoves to make them suitable for using them. This is important considering the relatively low incomes in rural areas. However, financial support for using straw pellets is only implemented in some provinces. Different from straw pellets, the prices for bioelectricity and biogas are the same as for fossil fuel as in these cases, subsidies are given to logistics companies and CRB producers.

Most logistics companies and CRB producers acknowledged the effectiveness of policies but considered that in their implementation delayed and inadequate subsidies were common. The amount of crop residues fluctuates depending on grain production and potential competition from other producers. The uncertainty of feedstock collection and purchase and technological limitations lead to higher production costs than for conventional fossil fuels. With respect to the cost of CRB production [58–60], we found that while financial support was granted when the projects were included in the list for subsidies, the approval process took time. Moreover, the subsidies come from national and local sources, whereby the local part depends on the available economic resources. Financial limitations are the most stringent constraint for developing CRB. Therefore, respondents suggest increasing financial support and simplifying processes in applying for subsidies.

Local civil servants and experts agreed that CRB development depended heavily on financial support and that small-scale CRB is suitable for rural development. Both national and local governments are faced with severe financial pressure in the promotion of CRB. To solve these financial issues, experts advised developing a carbon trading system. In that case, the financial contribution from fossil fuel use can be used to subsidize CRB development. In addition, small-scale CRB projects are more suitable for rural areas with their low population density. This finding was also reported in research on biomass gasification [10,59,60]. Especially in the northern part of China, crop residues, instead of coal, could be used as fuel for district heating and household stoves. Whether farmers use CRB or not, they pay the same amount for soil preparation after straw reuse for soil fertility or for collecting and baling crop residues; therefore, farmers are neutral with respect to the different modes of crop residue utilization. Thus, the preference of local civil servants is important in choosing between the different applications for comprehensively utilizing crop residues. It is important to note that fossil fuel projects are inefficient in rural areas

because reducing the cost of pollution emissions is usually only effective at a large scale. The collection and transport of crop residues for large-scale CRB projects, which may involve collecting crop residues over a distance much larger than 50 km [8], has a low economic efficiency.

### 4.3.2. Local Governments Favor Straw Return That Requires Lower Financial Support

We have shown that China's CRB policies are coherent at the level of objectives and instruments, but incoherent in the process of implementation. We also found that local governments preferred the reuse of straw for soil quality, followed by CRB and the other three applications. In their view, the targets for the CRB sector are too ambitious because its development mainly depends on national and local financial subsidies. The areas with a potential for developing CRB, where a large amount of crop residues is available, are major agricultural provinces with relatively few other economic resources. The subsidies needed for CRB increase the already existing financial stress for the local government. Moreover, the process of SR is much simpler than that of CRB. Crushing and mixing crop residues can usually be combined with harvest, tillage or soil preparation by using the same agricultural machines; therefore, incentives to purchase and operate agricultural machinery promote SR. Similar differences in policy coherence at different stages of policy making were also observed in the fields of biodiversity protection, waste treatment, fresh water protection and forestry [23,28]. In the context leading to this policy incoherence, the objectives of CRB development can be considered ambitious.

Compared with the withdrawal of significant investments in promoting CRB, the accountability for managing the ban on straw burning in the APPC Law is much stricter. The increase in public awareness of environmental protection and tighter accountability causes local governments to exert more pressure to monitor and take responsibility for managing this ban. In particular, the publication of real-time monitoring data provides concrete information to the public about their living environment. Convenient and anonymous ways to report environmental pollution make the public willing to join the environmental monitoring system. According to the China Environment Yearbook, the number of public complaints doubled from 2010 to 2015. Therefore, local governments prefer to choose SR because it is the most economic application for a comprehensive utilization of crop residues.

This observation can also explain the difference in policy outputs relating to crop residue reuse between Heilongjiang Province and Hunan Province. Compared with Hunan Province, Heilongjiang Province applies more instruments to prohibit the burning of straw and to promote crop residue utilization. The major crops in Heilongjiang are beans, corn, and rice, while the primary crop in Hunan is rice. Using corn for SR is more complicated than rice, and the average winter temperature in Heilongjiang hovers at approximately −20 degrees Celsius, leading to a harsh environment for reusing straw to improve soil quality. Therefore, the Hunan government received less pressure on crop residue disposal, while in Heilongjiang Province, burning straw occurred more frequently. The MEE satellite monitoring data showed more than 700 incidents of burning in 2016, and local supervising officials were punished for ineffective management. The Heilongjiang government has increased subsidies for promoting crop residue utilization formulated in national policies to utilize all crop residues and to further prohibit open air straw burning. The local government provided new subsidies for consuming straw pellets and constructing new CRB factories. All subsidies created financial pressure on the provincial budget, and this cannot be secured for a long time. The slowdown in CRB development was due to financial problems and competition among the producers of CRB.

Except for local financial pressure, the adoption of new technologies related to crop residue processing has received much pushback from farmers because of their low income. The presence of low incomes and changes in daily lives requires more policy support for promoting CRB in rural areas. Most Chinese farms are family scale, so it is challenging to promote agricultural mechanization. In addition, people are using coal for cooking or heating in rural areas and adopting biogas or other replacement leads to changes in cooking

routines in rural areas, which is a major obstacle for promoting CRB [10,61]. In addition to SR, forage is also a major competitor of CRB development. In summary, the different preferences relating to investments between national and local governments for utilizing crop residues resulted in not achieving the CRB policy objectives. Therefore, although horizontal coherence in policy documents is important, this is not sufficient because effective multisectoral policies require vertical coherence analysis and avoid implementation gaps.

*4.4. Policy Recommendations*

To achieve carbon neutrality, it is necessary to increase renewable energy use on a large scale. Although solar and wind power have been developed, both fluctuate in supply depending on the weather conditions. Therefore, bioenergy, including CRB, is essential to ensure a stable power supply and to mitigate GHG emissions. Based on the incoherence detected in this study, more debate is needed to discuss the different applications for comprehensively utilizing crop residues, especially between CRB and SR. A joint integrated mechanism for the crop residue utilization sector, instead of relying on the agricultural sector as the major governing department, may lead to a more horizontal coherent policy context and help to balance the different applications. Otherwise, the local government will tend to choose the application that needs the fewest financial resources. In addition, CRB is not available everywhere and depends on the local conditions, and the performance of local government needs to be paid more attention.

National guidance in choosing certain applications for utilizing crop residues across different areas is important to use local resources most efficiently, rather than a one-size-fits-all strategy. Policy proposals need to be based on a systematic assessment of local cropping systems, climate conditions (i.e., temperature and humidity), livestock development, and the availability of other renewable energy sources. Forage development depends on the scale of the local livestock sector. This finding is consistent with that of De Laurentis and Pearson [62] who highlight the regional physical resource endowment. Finally, other renewable energy resources and energy demands also influence the potential development of CRB. In places with a large potential for solar and wind power, other applications of crop residue utilization could be given preference. The more local conditions were involved in the policy process, the more coherent policies were and the easier support from local government was won.

Financial support largely affects the interest of stakeholders in CRB development. First, simplifying applications for subsidies for producing and consuming CRB could help to reduce local financial pressure and promote CRB development. Nevertheless, the complicated and time-consuming process of applying for subsidies would maintain the hesitation of many private actors to invest in crop residue logistics and CRB production. Second, subsidies for modifying household heating systems and purchasing CRB fuel help shape farmers' energy consumption practices and expand the CRB market. Third, incorporating CRB with a monitoring and verification system into a GHG emissions trading system including CRB would help to certify the environmental value of CRB. Many studies have shown that CRB production costs are higher than fossil fuels if environmental impacts are not considered. Once the CRB industries could obtain income from selling GHG emission permits, it would incentivize CRB production instead of relying on subsidies from governments that increase national and provincial financial stress. In summary, the more private actors participated in CRB development, the less financial pressure impeding the horizontal and vertical policy coherent the government would have.

In addition to the policy context, research is still needed to address the constraints of CRB development with respect to its technical and social aspects. Technological innovations are needed to lower the construction and operation costs of CRB. More attention should also be given to social aspects: what are the constraints for the consumers' adoption of CRB? For example, rural households' preference for biomass fuels could be affected by their economic ability. However, when people need to upgrade their stove system, straw pellets may be more suitable and widely introduced in these areas. Moreover, local governments could

recommend that farmers exchange their crop residues for CRB products, thereby promoting CRB adoption. All the above recommendations, policy related or not, do not only implicate China's CRB sector but may also inform other countries with high pressure of arable land and water resource to maintain food security, for example India and Bangladesh.

## 5. Conclusions

Our study has analyzed the horizontal coherence of multisectoral policies affecting CRB and other applications for recycling crop residue and the vertical coherence of CRB policy implementation in the cases of Heilongjiang and Hunan Provinces. We found that policies are horizontally coherent but vertically incoherent. The main conflict in crop residue application occurred between CRB and reusing straw for soil quality. The targets for CRB development in China are ambitious, particularly when considering the financial resources needed to achieve them. Local governments expressed a strong preference for SR because this requires less financial support while still meeting the goals of banning the burning of straw and of comprehensively utilizing crop residues. The preference at the provincial level for SR and decreased attention to the potential of CRB development also can be explained by the priority given to the cultivation and livestock sectors and the nature of the provincial agricultural department, which is the sector in charge. Moreover, we also provide policy recommendations based on the policy gaps we identified.

The empirical findings in this study provide an understanding of the impacts of trade-offs on policy effectiveness, especially for the nexus issues between multisectoral policies and different applications, such as competition between ecosystem services of biomass. As identified in China's CRB sector, the gaps between policy documents and their implementation clearly show that it is not enough to consider coherent objectives and instruments when aiming for an effective policy, especially in a cross-cutting sector. The vertical coherence analysis among policy implementation is essential for developing more effective policies. Given the current demand for renewable energy, this study contributes to building a framework for analyzing bioenergy and the related food-energy-water nexus within and outside of China. The issue of combined policy and technical aspects in the CRB industry could be usefully explored in further research because there are gaps in the CRB producing technology. In general, the results of this study show that combined horizontal and vertical policy coherence analysis is recommended to map out the effects, assess the effectiveness and efficiency of multisectoral policies, and further inform ways to adjust instruments and implementation strategies for cross-cutting issues.

**Author Contributions:** J.Z. performed the conceptualization, data collection and interviews in both provinces of Heilongjiang and Hunan. She also performed the integrated horizontal and vertical coherence to analyze the policy effectiveness in the case of China's CRB. Y.L. supervised these activities, and she is an agroclimatology expert responsible for giving insightful ideas about crop residues utilization in the policy recommendation section. P.O. and M.G. conceptualized and restructured the paper and supervised the research. All authors have read and agreed to the published version of the manuscript.

**Funding:** This work was funded by the Ministry of Agriculture and Rural Affairs of the People's Republic of China [Grant No. 13210352] and World Resources Institute [Grant No. G2585].

**Institutional Review Board Statement:** Not applicable.

**Informed Consent Statement:** Not applicable.

**Data Availability Statement:** Not applicable.

**Acknowledgments:** The authors want to thank the anonymous referees for their comments and suggestions. This work was funded by the projects of "Technologies for mitigation of GHG emissions and enhancement of carbon sequestration in agricultural sector and rural area of China" funded by the Ministry of Agriculture and Rural Affairs of the People's Republic of China [Grant No. 13210352] and the projcts of "GHG Reduction Pathways in planting Sector of China" funded by World Resources Institute [Grant No. G2585].

**Conflicts of Interest:** The authors declare no conflict of interest that could influence this study.

## Appendix A. Policy List

**Table A1.** National policies that influence the comprehensive utilization of crop residues.

| Policy Area and Reference | Policy | Sector |
|---|---|---|
| *GREEN AGRICULTURE* | | |
| SC 2007-1 | Vigorously Developing Modern Agriculture and the Construction of New Socialist Countryside | FG |
| SC 2010-59 | Accelerating the Transformation of Agricultural Development Mode in Northeast China | FG |
| SC 2012-4 | Plan for Agricultural Modernization (2011–2015) | FG |
| MOF 2015-31 | Guiding Opinion on Improving the Policies on Three Agricultural Subsidies | MOF |
| MOF 2016-26 | Effectively and Comprehensively Promoting the Reform of Three Agricultural Subsidies | MOF&MARA |
| MARA 2016-39 | Implementation of the Pilot Program on the Comprehensive Utilization of Crop Residues to Increase Soil Fertility | MARA&MOF |
| MARA 2016-6 | Launching the Pilot Program on Crop Rotation and Fallow | FG |
| MARA 2016-93 | Guidelines on the Ecological and Circular Agriculture Projects in Agricultural Comprehensive Development Zone | MARA |
| MARA 2016-28 | Measures for Monitoring of Arable Land in Pilot | MARA |
| SC 2016-58 | Plan for Agricultural Modernization (2016–2020) | FG |
| MARA 2017 | Pilot Program on Crop Rotation and Fallow in 2017 | MARA&MOF |
| MARA 2017-6 | The Implementation of the Five Major Actions on the Development of Green Agricultural | MARA |
| MARA 2017-24 | Ten Modes for Promoting Agricultural Utilization of Crop Residues | MARA |
| MARA 2017-9 | Action Plan for Comprehensive Utilization of Crop Residues in Northeast China | MARA |
| MARA 2017-11 | Effectively Conducting the Work of the Central Finance for the Agricultural Production Development Projects in 2017 | MARA&MOF |
| MARA 2017-3 | Outline of the Protection for Black Land in Northeast China (2017–2030) | FG |
| MARA 2017-12 | The Guiding Opinion on Accelerating the Development of Modern Animal Husbandry in the main grain-producing areas of Northeast China | MARA |
| SC 2017 | The Opinion Released to Innovate in Systems and Mechanisms for Green Agricultural Development | FG |
| MARA 2018 | Pilot Program on Crop Rotation and Fallow in 2018 | MARA&MOF |
| MARA 2018-13 | Effectively Conducting the Work of the Central Finance for the Agricultural Production Development Projects in 2018 | MARA&MOF |
| MARA 2019 | Pilot Program on Crop Rotation and Fallow in 2019 | MARA&MOF |
| MARA 2019-6 | Effectively Conducting the Work of the Central Finance for the Agricultural Production Development Projects in 2019 | MARA&MOF |
| MARA 2020 | Pilot Program on Crop Rotation and Fallow in 2020 | MARA&MOF |
| MARA 2020-2 | The Protective Action Plan for Black Land in Northeast China (2020–2025) | MARA&MOF |
| MARA 2020-3 | Effectively Conducting the Work of the Central Finance for the Agricultural Production Development Projects in 2020 | MARA&MOF |
| RE Law | Renewable Energy Law of the People's Republic of China | FG |
| NDRC 2007-44 | The Interim Measures for Allocation of Income from Surcharges on Renewable Energy Power Prices | NDRC |
| MARA 2007 | Plan for the Development of Agricultural Bioenergy during 2007–2015 | MARA |
| NDRC 2008-610 | The 11th Five-Year Plan for the Development of Renewable Energy | NDRC |
| RESF | Subsidization Funds for the Development of Cleaned Energy | MOF |
| NDRC 2010-1579 | Improving Price Subsidy Policy for Electricity Generated from Agricultural and Forestry Residues | NDRC |
| NEA 2012-216 | The 12th Five-Year Plan for the Development of Bioenergy | NEA |
| *RENEWABLE ENERGY* | | |
| NDRC 2006-7 | The Trial Measures for the Management of Prices and Allocation of Costs for Electricity Generated from Renewable Energy | NDRC |

**Table A1.** *Cont.*

| Policy Area and Reference | Policy | Sector |
|---|---|---|
| NDRC 2012-1207 | The 12th Five-Year Plan for the Development of Renewable Energy | NDRC |
| SC 2014-31 | The Program of Action for the Energy Development Strategy (2014–2020) | FG |
| NEA 2016-291 | The 13th Five-Year Plan for the Development of Bioenergy | NEA |
| NDRC 2016-2619 | The 13th Five-Year Plan for the Development of Renewable Energy | NDRC |
| NDRC 2017-2143 | Guiding Opinion on Carrying Out the Construction of Straw Gasification and Clean Energy Utilization Projects | NDRC |
| **ENVIRONMENTAL PROTECTION & RESOURCE RECYCLING** | | |
| MEE 1999-98 | Measures for Straw Burning Ban and the Comprehensive Utilization of Crop Residues | MEE |
| APPC Law | Atmospheric Pollution Prevention and Control Law of the People's Republic of China (since the 2000 version) | FG |
| MEE 2002-56 | The 10th Five-Year Plan for the Protection of Ecological Environment | MEE |
| MEE 2003-78 | Strengthening the Work on Straw Burning Ban Management and the Comprehensive Utilization of Crop Residues | MEE |
| TP Catalog | The Catalog of Corporate Tax Reductions for the Comprehensive Utilization of Resources | MOF/NDRC |
| SC 2005-21 | The Near-term Key Work in Building a conservation-minded Society | FG |
| SC 2007-23 | The 11th Five-Year Plan for the Bio-Industry Development | FG |
| SC 2008-17 | National Plan for the China's Responses to Climate Change | FG |
| SC 2008-63 | The Opinion on Strengthening Rural Environmental Protection Work | FG |
| SC 2008-105 | The Opinion on Accelerating the Comprehensive Utilization of Crop Residues | FG |
| MEE 2008-22 | Further Strengthening the Work on Straw Burning Ban Management | MEE |
| SC 2010-33 | Promoting Joint Prevention and Control of Air Pollution to Improve Air Quality | FG |
| NDRC 2011-2615 | Implementation Plan for Comprehensive Utilization of Crop Residues during the 12th Five-Year | NDRC |
| SC 2012-28 | The 12th Five-Year Plan for the Development of the Strategic Emerging Industries | FG |
| SC 2013-5 | The Circular Economy Development Strategy and Near-Term Action Plan | FG |
| SC 2013-65 | Notice of the State Council on Issuing the Bio-Industry Development Plan | FG |
| NDRC 2013-930 | Strengthening Comprehensive Utilization of Crop Residues and Straw Burning Ban Management | NDRC |
| SC 2014-21 | The Measures for Evaluating the Implementation of the Air Pollution Prevention and Control Action Plan (for Trial Implementation) | FG |
| SC 2014-25 | The Guiding Opinion of the General Office of the State Council on Improving Rural Living Environment | FG |
| NDRC 2014-2231 | Work Plan on Comprehensive Utilization of Crop Residues and Straw Burning Ban Management in Beijing, Tianjin, Hebei, and Curbing Area | NDRC |
| NDRC 2015-2651 | Further Accelerating Comprehensive Utilization of Crop Residues and Straw Burning Ban Management | NDRC |
| MARA 2016-90 | The Work Plan for the Pilot Program of Agricultural Resource Utilization | MARA |
| NDRC 2016-2504 | Guiding on the Implementation Plan for the Comprehensive Utilization of Crop Residues during the 13th Five-Year | NDRC |
| SC 2016-66 | The 13th Five-Year Plan for the Protection of Ecological Environment | FG |
| SC 2016-67 | The 13th Five-Year Plan for the Development of the Strategic Emerging Industries | FG |
| NDRC 2016-2665 | The 13th Five-Year Plan for the Bio-Industry Development | NDRC |
| SC 2017-74 | Comprehensive Five-Year (2016–2020) Plan for Energy Conservation and Emission Reduction Issued | FG |
| EPT Law | Environmental Protection Tax Law of the People's Republic of China[1] | FG |
| SC 2018-22 | The Three-Year Action Plan for the Bule Sky | FG |
| MARA 2018-4 | The Opinion regarding further promoting Ecological Environment Protection | MARA |
| MEE 2018-143 | Action Plan for the Battle of Agricultural and Rural Pollution Control | MEE |
| SC 2018-128 | The Work Plan for the Pilot Program of "Zero-Waste City" Building | FG |

## Appendix B. Policy Instruments

**Table A2.** Policy Impacts on Crop Residue Utilization.

| Policy Outputs | Type | Reference * | CRB | SR | Others |
|---|---|---|---|---|---|
| **CROP RESIDUE PRODUCTION** | | | | | |
| Financing fallow for sustainable agricultural development | Eco. | MARA 2018-4; SC 2017; SC 2016-66; MARA 2016-6; MARA 2016-28; MARA 2020-3; | (−) | (−) | (−) |
| Prohibition on burning straw in the open air, including fines, information, developing regulations, etc. | Reg. | MARA 2017-6; MARA 2016-39; SC 2010-33; SC 2018-22; SC 2017-74; SC 2016-66; SC 2008-105; NDRC 2015-2651; NDRC 2016-2504; MARA 2017-9; MEE 2008-22; MEE 2018-143; APPC Law (since 2000 version); SC 2014-21; | (+) | (+) | (+) |
| **CROP RESIDUE LOGISTICS** | | | | | |
| Encourage private sectors to engage in crop residue logistics by financing the construction of crop residue infrastructure, i.e., warehouse and land use | Gov. Services | MARA 2016-90; MARA 2016-39; SC 2018-128; SC 2008-105; NDRC 2015-2651; MARA 2016-93; NDRC 2016-2504; MARA 2017-9; APPC Law; | (+) | (?) | (+) |
| Financing the collection and transport of crop residues from arable land | Eco. | MARA 2020-3; APPC Law; | (+) | (?) | (+) |
| **COMPREHENSIVE UTILIZATION OF CROP RESIDUES** | | | | | |
| Increasing the overall utilization ratio of crop residues | Reg. | MARA 2017-6; MARA 2016-90; MARA 2016-39; SC 2017; SC 2018-128; SC 2018-22; SC 2017-74; SC 2013-5; SC 2016-58; SC 2016-66; NDRC 2015-2651; MARA 2016-93; NDRC 2016-2504; MARA 2017-9; | + | + | + |
| Financing innovations in manufacturing equipment to increase efficiency in crop residue processing | Gov. Services | MARA 2016-90; SC 2010-59; SC 2016-67; SC 2008-105; NDRC 2015-2651; NDRC 2016-2504; MARA 2017-9; NDRC 2016-2665; RESF; SC 2007-1; | + | + | + |
| Encourage public participation by involving the public in the implementation and assessment of policies | Info. | SC 2017; SC 2014-25; SC 2018-22; SC 2017-74; SC 2016-58; NDRC 2015-2651; NDRC 2016-2504; MARA 2017-9; MARA 2020-3; NDRC 2016-2619; SC 2014-21; | + | + | + |

**Table A2.** *Cont.*

| Policy Outputs | Type | Reference * | CRB | SR | Others |
|---|---|---|---|---|---|
| Promoting utilization by providing technical support | Info. | MARA 2018-4; MARA 2017-12; MARA 2016-39; SC 2013-5; SC 2008-105; NDRC 2015-2651; NDRC 2016-2504; MARA 2017-9; MARA 2017-24; MARA 2016-28; MARA 2017-3; | + | + | + |
| Financing the applications for crop residue utilization, including subsidies and tax reductions | Eco. | MARA 2017-12; SC 2018-128; SC 2008-105; NDRC 2015-2651; NDRC 2016-2665; TP catalog; APPC Law; | + | + | + |
| Environmental tax, or decreasing financial support for fossil fuel-based products/chemical fertilizer | Eco. | SC 2018-128; SC 2017-74; SC 2014-31; NDRC 2016-2619; EPT Law; | (+) | (+) | (+) |
| **Crop Residue Utilization for Bioenergy** | | | | | |
| More bioenergy/CRB is needed to reach the GHG emission reduction target | Reg. | SC 2016-67; NEA 2016-291; SC 2014-31; NDRC 2016-2619; NDRC 2016-2665; | + | (−) | (−) |
| Financing CRB producers, i.e., tax reduction, subsidies for factory construction, and guaranteeing the purchase of CRB products | Reg. & Eco. | SC 2018-22; NDRC 2015-2651; RE Law; NDRC 2010-1579; NDRC 2007-44; NDRC 2016-2619; NDRC 2006-7 | + | (−) | (−) |
| Constructing infrastructures for transporting CRB products, i.e., power grid and biogas pipes | Gov. Services | SC 2018-22; MARA 2016-93; NEA 2016-291; RE Law; MEE 2018-143; NDRC 2016-2619; | + | (?) | (?) |
| Financing CRB users, e.g., subsidy for straw pellets stoves or for switching from coal to straw pellets, etc. | Eco. | SC 2014-25; NDRC 2015-2651; NEA 2016-291; NDRC 2017-2143; NDRC 2016-2665; | + | (?) | (?) |
| **Crop Residue Utilization for Soil Improvement** | | | | | |
| Crop residues should be returned to arable land to increase soil fertility * | Reg. | MARA 2016-39; SC 2016-58; SC 2008-105; NDRC 2016-2504; MEE 2018-143; MARA 2020-2; MARA 2017-3; MOF 2016-26; APPC Law; SC 2007-1; | (−) | + | (−) |
| Financing straw return to arable land, including subsidies for purchasing agricultural machines and straw return | Eco. | MARA 2016-39; SC 2016-58; SC 2008-105; NDRC 2016-2504; MEE 2018-143; MARA 2020-2; MARA 2017-3; MOF 2016-26; APPC Law; SC 2007-1; | (−) | + | (−) |

Note: * The policy documents for each abbreviation are shown in Table A1 in Appendix A. "+" strengthening or positive impacts, "−" weakening or negative impacts, and "?" unknown impacts that could be positive, negative or neutral impacts. The parentheses in the effects indicate that the effect of the output is indirect. Eco. denotes economic instruments, Reg. denotes regulatory instruments, Info. denotes informative instruments, Gov. Services denote the government public goods and services provision.

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
