# Peer review of "Bioenergy versus Soil Improvement: Policy Coherence and Implementation Gaps in Crop Residue-Based Bioenergy Development in China"

_water, doi:10.3390/w14213527_

Round 1

Reviewer 1 Report

Attached as a document.

Author Response

Please see the attached response letter.

Reviewer 2 Report

This work analyzes the horizontal coherence of national multisectoral policies, including agricultural, energy and environmental scopes, on CRB (crop residue-based bioenergy) performance in China and the vertical coherence among policy implementations. The authors have done a detailed analysis of all policies related to this problem and have been able to reach important conclusions to improve in the future. In my opinion, the work is ready to be published.

Minor correction: In figure 5, correct “negetive” and “environental”.

Author Response

Comment 1: Minor correction: In figure 5, correct “negetive” and “environental”.

Response: Thanks for your comments. I have edited the content in Figure 5 and updated the manuscript. Please see the new figure in the attachment.
